# Stability of Fractional-Order Quasi-Linear Impulsive Integro-Differential Systems with Multiple Delays

**Mathiyalagan Kalidass** [1,*] **, Shengda Zeng** [2,*] **and Mehmet Yavuz** [3]

1   Department of Mathematics, Bharathiar University, Coimbatore 641 046, India; kmathimath@gmail.com
2   Guangxi Colleges and Universities Key Laboratory of Complex System Optimization and Big Data Processing, Yulin Normal University, Yulin 537000, China
3   Department of Mathematics and Computer Sciences, Faculty of Science, Necmettin Erbakan University, Konya 42090, Türkiye; mehmetyavuz@erbakan.edu.tr
*   Correspondence: kmathimath@gmail.com (M.K.); shengdazeng@ylu.edu.cn (S.Z.)

**Abstract:** In this paper, some novel conditions for the stability results for a class of fractional-order quasi-linear impulsive integro-differential systems with multiple delays is discussed. First, the existence and uniqueness of mild solutions for the considered system is discussed using contraction mapping theorem. Then, novel conditions for Mittag–Leffler stability (MLS) of the considered system are established by using well known mathematical techniques, and further, the two corollaries are deduced, which still gives some new results. Finally, an example is given to illustrate the applications of the results.

**Keywords:** fractional-order system; quasi-linear system; impulse; integro-differential equation; stability; multiple delays





## 1. Introduction

Differential equations involving an arbitrary non-integer order are often used as excellent tools for describing many dynamical processes because they have nonlocal properties and weakly singular kernels; for more details, see [1–3]. Most the investigations show that non-integer order calculus is more suitable and has accuracy when describing various physical systems in areas such as mechanical systems, electro-chemistry, biological systems and diffusion processes; see, for instance, [4–8]. Further, as pointed out in [9], the fractional-order derivative provides fundamental and general computation ability for efficient information processing and stimulus anticipation for real models. Usually, systems with nonlocal conditions are generalizations of local nonlinear boundary conditions, which gives better approximations in some physical problems [10]. Further, the quasi-linear integro-differential equations have occurred during the study of the nonlinear behavior of elastic strings and other areas of physics. Many interesting results on various forms of systems, including fractional-order, quasi-linear, integro-differential and non-local systems, are found in [11–15] and references therein.

On the other hand, impulses in differential equations reflect the dynamics of real world problems with unexpected discontinuities and rapid changes at certain instants, such as blood flows, heart beats and so on [16]. Impulsive behavior often exists in many real world systems. Fundamentally, the impulses are samples of state variables of a controlled system at discrete moments. These effects most often occur in pharmacokinetics, the radiation of electromagnetic waves, nanoelectronics, etc. [17]. There are number of interesting research papers on impulsive differential equations found in the literature; see [18–20] and references therein. The piecewise-continuous solutions for the impulsive Cauchy problem and impulsive boundary-value problem were studied in [21]. The existence and finite-time stability of an impulsive fractional-order system (FOS) using Gronwall inequality involving Hadamard-type singular kernel has been investigated in [20]. Wang et al. [19] derived

the finite-time stability of impulsive fractional-order delayed systems using generalized Bellman-Gronwall's inequality.

The sufficient conditions for MLS and uniform asymptotic stability of nonlinear impulsive FOSs were obtained in [22]. The MLS of nonlinear FOSs with impulses has been analyzed in [23]. The MLS for impulsive FOSs with instantaneous and non-instantaneous impulses were studied in [24]. The MLS of a nonlinear FOS was studied in [25] by extending the Lyapunov direct method. The MLS for a coupled system of FOSs with impulses was investigated in [26]. The MLS for nonlinear fractional neutral singular systems were obtained in [27]. The finite time stability of delayed FOSs by Mittag–Leffler functions was analyzed in [28]. An MLS estimator for a nonlinear FOS using a linear quadratic regulator approach was studied in [29]. Many problems in viscoelasticity, acoustics, populations dynamics, electromagnetics, hydrology, chemical reactions and other areas can be modeled by fractional integro-differential equations; see [30–32] and references therein. For example, take the the nonlinear oscillation of earthquake model, fluid-dynamic traffic model, second-grade fluid model, circulant Halvorsen system, susceptible-infected-recovered epidemic model with a fractional derivative and many other recent developments in the description of anomalous by fractional dynamics; see [33–36].

The stability of dynamical systems is an essential one in the qualitative theory of dynamical systems, as it addresses the system trajectories under small perturbations of initial conditions. The stability analysis of FOS is more difficult than the classical ones because the fractional-order derivative is nonlocal and has infirm singular kernels [37,38]. In the literature, the concepts of the stability analysis of impulsive FOS are studied by various approaches. Among them, MLS is more useful in FOSs because the Mittag–Leffler functions are commonly used in fractional calculus, which generally features power-law convergence. Thus, in this paper we made an attempt to study MLS analysis for quasi-linear impulsive FOS with multiple time delays. Recently, many authors focused on the various types of stability analysis for FOS; for example, the q-MLS and direct Lyapunov method for q-FOS is discussed in [39]. The Mittag–Leffler input stability of FOSs with exogenous disturbances using the Lyapunov characterization is studied in [40]. Li et al. [41] proposed the MLS using the fractional Lyapunov direct method.

However, there are few results available for the MLS of FOS with impulse effects that could not be suitable for FOSs of quasi-linear type with multiple time delays. To the best of our knowledge, the Mittag–Leffler stability of FOSs has not been fully investigated, which motivated our present study. Thus, in this study the existence and uniqueness of solutions and MLS analysis of the impulsive quasi-linear FOS with multiple time delays are established using the well-known fixed point theorems and Mittag–Leffler approach. Further, the main contribution of this paper lies in deriving new stability conditions for the fractional-order quasi-linear system with nonlocal conditions, multiple time delays and impulses. Novel conditions for the Mittag–Leffler stability of FOSs is established. The existence and uniqueness of mild solutions for the FOS are discussed with help of the contraction mapping principle. Finally, an example is provided to show the applicability of the results.

## 2. Problem Description

Consider the fractional model given by

$$D^\beta \mathfrak{z}(t) + A(t, \mathfrak{z}(t))\mathfrak{z}(t) = \mathfrak{f}(t, \mathfrak{z}(t), \mathfrak{z}(\tau(t))) + \int_0^t \mathfrak{g}(t, \alpha, \mathfrak{z}(\alpha), \mathfrak{z}(\delta(\alpha)))d\alpha,$$

$$\mathfrak{z}(0) + \mathfrak{h}(\mathfrak{z}) = \mathfrak{z}_0, \tag{1}$$

$$\Delta\mathfrak{z}(t_k) = I_k(\mathfrak{z}(t_k)),$$

in Banach space $X$, $0 < \beta \leq 1$, $t \in J = [0, T]$, $\mathfrak{z}_0 \in X$, $k = 1, 2, \ldots, \mathfrak{m}$ and $0 < t_1 < t_2 < \ldots < t_\mathfrak{m} < T$. Assume $-A(t, \mathfrak{z}(t))$ is a closed linear operator defined on a dense domain $D(A)$ in $X$ into $X$ such that $D(A)$ is independent of $t$, and it generates an evolution operator in $X$.

Let $\mathfrak{z} \in PC(J, X)$ be continuous at $t \neq t_k$ and left continuous at $t = t_k$; in addition, right limit $\mathfrak{z}(t_k^+)$ exists for $k = 1, 2, \ldots, \mathfrak{m}$. Clearly $PC(J, X)$ is a Banach space with the norm $\|\mathfrak{z}\|_{PC} = \sup_{t \in J} \|\mathfrak{z}(t)\|$. Additionally, $\mathfrak{z}(\tau) = (\mathfrak{z}(\tau_1), \mathfrak{z}(\tau_2), \ldots, \mathfrak{z}(\tau_r))$ and $\mathfrak{z}(\delta) = (\mathfrak{z}(\delta_1), \mathfrak{z}(\delta_2), \ldots, \mathfrak{z}(\delta_\kappa))$ are multiple time-delays. The functions $\mathfrak{f}$, $\mathfrak{g}$ and $\mathfrak{h}$ are nonlinear in nature, satisfying:

$(H_1)$ function $\mathfrak{f} \colon J \times X^{r+1} \to X$ is continuous, and there exist positive constants $\mathfrak{f}_1$, $\mathfrak{f}_2$ such that

$$\|\mathfrak{f}(t, \mathfrak{z}_1, \mathfrak{z}_2, \ldots, \mathfrak{z}_{r+1}) - \mathfrak{f}(t, \tilde{\mathfrak{z}}_1, \tilde{\mathfrak{z}}_2, \ldots, \tilde{\mathfrak{z}}_{r+1})\|_X \leq \mathfrak{f}_1 \sum_{p=1}^{r+1} \|\mathfrak{z}_p - \tilde{\mathfrak{z}}_p\|_X,$$

$\mathfrak{z}_p, \tilde{\mathfrak{z}}_p \in X$ and $\mathfrak{f}_2 = \max_{t \in J} \|\mathfrak{f}(t, 0, \ldots, 0)\|_X$.

$(H_2)$ function $\mathfrak{g} \colon \Lambda \times X^{\kappa+1} \to X$ is continuous, and there exist positive constants $\mathfrak{g}_1$, $\mathfrak{g}_2$ such that

$$\int_0^t \|\mathfrak{g}(t, \alpha, \mathfrak{z}_1, \mathfrak{z}_2, \ldots, \mathfrak{z}_{\kappa+1}) - \mathfrak{g}(t, \alpha, \tilde{\mathfrak{z}}_1, \tilde{\mathfrak{z}}_2, \ldots, \tilde{\mathfrak{z}}_{\kappa+1})\|_X d\alpha$$

$$\leq \mathfrak{g}_1 \sum_{q=1}^{\kappa+1} \|\mathfrak{z}_q - \tilde{\mathfrak{z}}_q\|_X, \ \mathfrak{z}_q, \tilde{\mathfrak{z}}_q \in X,$$

$$\mathfrak{g}_2 = \max \left\{ \int_0^t \|\mathfrak{g}(t, \alpha, 0, \ldots, 0)\|_X d\alpha \colon (t, \alpha) \in \Lambda \right\}.$$

$(H_3)$ $\tau_p$ and $\delta_q \colon J \to J$ are bijective and absolutely continuous, and there exist constants $c_p$ and $b_q$ such that $\tau_p'(t) \geq c_p$ and $\delta_q'(t) \geq b_q$, respectively, for $t \in J$ and $\Lambda = \{(t, \theta), 0 \leq \theta \leq t \leq T\}$.

$(H_4)$ Let $\Omega$ be a subset of $X$, and $\mathfrak{h} \colon PC(J, \Omega) \to Y$ is Lipschitz continuous in $X$ and bounded in $Y$; i.e., there exist positive constants $\mathfrak{h}_1$, $\mathfrak{h}_2$ such that

$$\|\mathfrak{h}(\mathfrak{z})\|_Y \leq \mathfrak{h}_1 \text{ and } \|\mathfrak{h}(\mathfrak{z}) - \mathfrak{h}(\tilde{\mathfrak{z}})\|_Y \leq \mathfrak{h}_2 \max_{t \in J} \|\mathfrak{z} - \tilde{\mathfrak{z}}\|_{PC}, \ \mathfrak{z}, \tilde{\mathfrak{z}} \in PC(J, X).$$

$(H_5)$ $\mathfrak{I}_k \colon X \to X$ are continuous and there exist constants $\mathfrak{l} > 0$, such that $\|\mathfrak{I}_k(\mathfrak{z}) - \mathfrak{I}_k(\tilde{\mathfrak{z}})\| \leq \mathfrak{l}\|\mathfrak{z} - \tilde{\mathfrak{z}}\|$, $\mathfrak{z}, \tilde{\mathfrak{z}} \in X$, where $k = 1, 2, 3, \ldots, \mathfrak{m}$.

*2.1. Preliminaries*

Let $X$ and $Y$ be two Banach spaces such that $Y$ is densely and continuously embedded in $X$. For Banach space, the norm of $X$ is denoted by $\|.\|_X$. The space of all bounded linear operators from $X$ to $Y$ is denoted by $\mathcal{B}(X, Y)$, and $\mathcal{B}(X, X)$ is written as $\mathcal{B}(X)$.

Now we recall some basic definitions and lemmas which will be useful in the main results.

**Definition 1.** *A two-parameter family of bounded linear operators $\mathfrak{U}(t, \Theta)$ and $0 \leq \Theta \leq t \leq T$, on $X$, is called an evolution system if the following two conditions are satisfied:*
*(1)* $\mathfrak{U}(t, t) = I$, $\mathfrak{U}(t, r)\mathfrak{U}(r, \Theta) = \mathfrak{U}(t, \Theta)$ *for* $0 \leq \Theta \leq r \leq t \leq T$,
*(2)* $(t, \Theta) \to \mathfrak{U}(t, \Theta)$ *is strongly continuous for* $0 \leq \Theta \leq t \leq T$.

Let $E$ be the Banach space formed from domain $D(A)$ with the graph norm. Since $-A(t)$ is a closed operator, it follows that $-A(t)$ is in the set of bounded operators from $E$ to $X$.

**Definition 2.** *A resolvent operator for* (1) *is a bounded operator-valued function $R_{\mathfrak{z}}(t, \Theta) \in \mathcal{B}(X)$, $0 \leq \Theta \leq t \leq T$, the space of bounded linear operator on $X$, having the following properties:*
- *$R_{\mathfrak{z}}(t, \Theta)$ is strongly continuous in $\Theta$ and $t$, $R_{\mathfrak{z}}(\Theta, \Theta) = I$, $0 \leq \Theta \leq T$, $\|R_{\mathfrak{z}}(t, \Theta)\| \leq Y e^{N(t-\Theta)}$ for some constants $Y$ and $N$.*
- *$R_{\mathfrak{z}}(t, \Theta)E \subset E$, $R_{\mathfrak{z}}(t, \Theta)$ is strongly continuous in $\Theta$ and $t$ on $E$.*

- *For $x \in X$, $R_{\mathfrak{z}}(t, \Theta)x$ is continuously differentiable for $\Theta \in [0, T]$ and $\frac{\partial R_{\mathfrak{z}}}{\partial \Theta}(t, \Theta)x = R_{\mathfrak{z}}(t, \Theta)A(\Theta, \mathfrak{z}(\Theta))x$.*
- *For $x \in X$ and $\Theta \in [0, T]$, $R_{\mathfrak{z}}(t, \Theta)x$ is continuously differentiable for $t \in [\Theta, T]$ and*
$$\frac{\partial R_{\mathfrak{z}}}{\partial t}(t, \Theta)x = -A(t, \mathfrak{z}(t))R_{\mathfrak{z}}(t, \Theta)x,$$

*with $\frac{\partial R_{\mathfrak{z}}}{\partial \Theta}(t, \Theta)x$ and $\frac{\partial R_{\mathfrak{z}}}{\partial t}(t, \Theta)x$ are strongly continuous on $0 \leq \Theta \leq t \leq T$. Further, $R_{\mathfrak{z}}(t, \Theta)$ can be extracted from the evolution operator of the generator $-A(t, \mathfrak{z})$. The resolvent operator is similar to the evolution operator for non-autonomous systems in a Banach space.*

The Mittag–Leffler function (MLF) in one parameter is defined by $E_{\beta}(z) = \sum\limits_{n=0}^{\infty} \frac{z^n}{\Gamma(\beta n + 1)}$ where $\beta > 0$ and MLF in two parameters is $E_{\beta_1, \beta_2}(z) = \sum\limits_{n=0}^{\infty} \frac{z^n}{\Gamma(\beta_1 n + \beta_2)}$ where $\beta_1 > 0$, $\beta_2 > 0$ and $z \in C$. Additionally, for $\beta_2 = 1$, $E_{\beta_1}(z) = E_{\beta_1, 1}(z)$ and $E_{1,1}(z) = e^z$. Further, the Laplace transform of MLF in two parameters is $L\{t^{\beta_2 - 1}E_{\beta_1, \beta_2}(-\gamma t^{\beta})\} = \frac{s^{\beta_1 - \beta_2}}{s^{\beta_1} + \gamma}$ for $t \geq 0$, where $\gamma, s \in \mathbb{R}$.

**Lemma 1** ([21])**.** *Let $\beta \in (0, 1)$ and $\mathfrak{f} \colon J \to R$ be continuous. A function $\mathfrak{z}(t) \in C(J, R)$ given by*

$$\mathfrak{z}(t) = \mathfrak{z}_0 - \frac{1}{\Gamma(\beta)} \int_0^a (a - \alpha)^{\beta - 1} \mathfrak{f}(\alpha) d\alpha + \frac{1}{\Gamma(\beta)} \int_0^t (t - \alpha)^{\beta - 1} \mathfrak{f}(\alpha) d\alpha,$$

*is the only solution of the fractional Cauchy problem $^cD_t^{\beta} = \mathfrak{f}(t)$ for all $t \in J$, $\mathfrak{z}(a) = \mathfrak{z}_0$, where $a > 0$.*

**Lemma 2** ([10])**.** *Let $R_{\mathfrak{z}}(t, \Theta)$ and $R_{\tilde{\mathfrak{z}}}(t, \Theta)$ be the resolvent operators for system (1). There exists a constant $c > 0$ such that*

$$\|R_{\mathfrak{z}}(t, \Theta)W - R_{\tilde{\mathfrak{z}}}(t, \Theta)W\| \leq c\|W\|_Y \int_{\Theta}^t \|\mathfrak{z}(\sigma) - \tilde{\mathfrak{z}}(\sigma)\| d\sigma,$$

*for every $\mathfrak{z}, \tilde{\mathfrak{z}} \in PC(J, X)$ and every $W \in Y$.*

Let $S_{\lambda} = \{\mathfrak{z} \colon \mathfrak{z} \in PC(J, X), \mathfrak{z}(0) + \mathfrak{h}(\mathfrak{z}) = \mathfrak{z}_0, \Delta\mathfrak{z}(t_k) = \mathfrak{I}_k(\mathfrak{z}(t_k)), \|\mathfrak{z}\| \leq \lambda\}$, for $t \in J$, $\lambda > 0$, $\mathfrak{z}_0 \in X$ and $k = 1, 2, 3, \ldots, \mathfrak{m}$.

**Lemma 3** ([12])**.** *For*

$$\phi(t) = \frac{1}{\Gamma(\beta)} \int_0^t (t - \alpha)^{\beta - 1} [\mathfrak{f}(\alpha, \mathfrak{z}(\alpha), \mathfrak{z}(\tau(\alpha))) + \int_0^{\alpha} \mathfrak{g}(\alpha, \eta, \mathfrak{z}(\eta), \mathfrak{z}(\delta(\eta))) d\eta] d\alpha,$$

*there exists a constant $\theta$ such that $\|\phi(t)\|_Y \leq \theta$ holds.*

*2.2. Existence and Uniqueness*

Before presenting the stability results, we discuss the existence and uniqueness of mild solutions for the FOS (1).

**Theorem 1.** *Let $-A(t, \mathfrak{z}(t))$ generate the resolvent operator $\|R_{\mathfrak{z}}(t, \Theta)\| \leq Y e^{N(t - \Theta)}$ with $Y_0 = \max \|R_{\mathfrak{z}}(t, \Theta)\|_Y$ for all $0 \leq \Theta \leq t \leq T$, $\mathfrak{z} \in \Omega$, and the conditions $(H_1)$–$(H_5)$ hold. If there exist positive constants $\lambda_1, \lambda_2, \lambda_3 \in (0, \frac{\lambda}{3}]$ and $\rho_1, \rho_2, \rho_3 \in [0, \frac{1}{3})$ such that*

$\lambda_1 = Y_0\|u_0\|_Y + Y_0\mathfrak{h}_1$, $\lambda_2 = Y_0\nu$, $\lambda_3 = Y_0\mathfrak{ml}\lambda$ *and*

$$
\begin{aligned}
\rho_1 &= cT\|\mathfrak{z}_0\|_Y + \mathfrak{h}_1 cT + Y_0\mathfrak{h}_2, \\
\rho_2 &= cT\nu + Y_0\frac{T^\beta}{\Gamma(1+\beta)}\Big[\mathfrak{f}_1\Big(1+\frac{1}{c_1}+\ldots+\frac{1}{c_r}\Big) + \mathfrak{g}_1\Big(1+\frac{1}{b_1}+\ldots+\frac{1}{b_\kappa}\Big)\Big], \\
\rho_3 &= cT\mathfrak{ml}\lambda + Y_0\mathfrak{ml}, \; \text{where} \; \sum_{k=1}^{\mathfrak{m}}\mathfrak{l} = \mathfrak{ml}, \\
\xi &= \frac{T^\beta}{\Gamma(1+\beta)}\Big[\mathfrak{f}_1\Big(\frac{1}{c_1}+\ldots+\frac{1}{c_r}\Big) + \mathfrak{g}_1\Big(\frac{1}{b_1}+\ldots+\frac{1}{b_\kappa}\Big)\Big], \\
\nu &= \frac{T^\beta}{\Gamma(1+\beta)}\lambda(\mathfrak{g}_1+\mathfrak{f}_1) + \xi\lambda + \frac{T^\beta}{\Gamma(1+\beta)}(\mathfrak{g}_2+\mathfrak{f}_2)
\end{aligned}
$$

*are satisfied, then the system* (1) *has a unique mild solution*

$$
\begin{aligned}
\mathfrak{z}(t) &= R_{\mathfrak{z}}(t,0)\mathfrak{z}_0 - R_{\mathfrak{z}}(t,0)\mathfrak{h}(\mathfrak{z}) + \frac{1}{\Gamma(\beta)}\int_0^t (t-\alpha)^{\beta-1}R_{\mathfrak{z}}(t,\Theta)[\mathfrak{f}(\alpha,\mathfrak{z}(\alpha),\mathfrak{z}(\tau(\alpha))) \\
&\quad + \int_0^\alpha \mathfrak{g}(\alpha,\eta,\mathfrak{z}(\eta),\mathfrak{z}(\delta(\eta)))\mathrm{d}\eta]\mathrm{d}\alpha + \sum_{0<t_k<t} R_{\mathfrak{z}}(t,t_k)\mathfrak{I}_k(\mathfrak{z}(t_k))
\end{aligned} \tag{2}
$$

*on J for all* $\mathfrak{z}_0 \in X$.

By contraction mapping theorem, the unique mild solution of the form (2) for system (1) can be easily derived; for detailed proof, one can refer to [12].

**Remark 1.** *It is noted that in addition to the Assumptions* $(H_1)$–$(H_5)$, *if Y is reflexive and the functions* $\mathfrak{f}$ *and* $\mathfrak{g}$ *are uniformly Hölder continuous, then the system* (1) *has a unique classical solution similar to* (2) *on J.*

## 3. Stability Results

In this section, we prove the Mittag–Leffler stability of the considered system.

**Definition 3.** *The mild solution of system* (1) *is said to be Mittag–Leffler stable if there exists a constant* $\beta \in (0,1)$ *and positive constants a, b, M and μ such that the solution* $\mathfrak{z}(t)$ *of system* (1) *satisfies*

$$
\|\mathfrak{z}(t)\| \leq M\|\mathfrak{z}_0\|^b \Big(E_\beta(-\mu(t-t_0)^\beta)\Big)^a, \quad t \geq 0.
$$

**Theorem 2.** *Let* $-A(t,\mathfrak{z}(t))$ *generate the bounded resolvent operator* $\|R_{\mathfrak{z}}(t,\Theta)\| \leq Ye^{N(t-\Theta)}$ *with* $Y_0 = \max\|R_{\mathfrak{z}}(t,\Theta)\|_Y$ *for all* $0 \leq \Theta \leq t \leq T$, $\mathfrak{z} \in \Omega$, *and the conditions* $(H_1)$–$(H_5)$ *hold. If there exist constants* $\mathfrak{f}_1$, $\mathfrak{g}_1$, $\mathfrak{h}_1$, *the mild solution of system* (1) *satisfies*

$$
\|\mathfrak{z}(t)\| \leq (1/\vartheta)Y_0(\|\mathfrak{z}_0\| + \mathfrak{h}_1)E_\beta\Big(\mu t^\beta\Big), \quad \forall t \in J, \tag{3}
$$

*where* $\vartheta = (1 - Y_0\mathfrak{ml})$ *and* $\mu = \frac{Y_0(\mathfrak{f}_1+\mathfrak{g}_1)}{\vartheta}$, *so the system* (1) *is Mittag–Leffler stable.*

**Proof.** Consider the mild solution of the system (1) of (2). Taking the norm on both sides, one can have

$$
\begin{aligned}
\|\mathfrak{z}(t)\| \;\leq\; & \|R_{\mathfrak{z}}(t,0)\|\,\|\mathfrak{z}_0\| + \|R_{\mathfrak{z}}(t,0)\|\,\|\mathfrak{h}(\mathfrak{z})\| \\
& + \frac{1}{\Gamma(\beta)} \int_0^t (t-\alpha)^{\beta-1}\|R_{\mathfrak{z}}(t,\Theta)\|\,\|\mathfrak{f}(\alpha,\mathfrak{z}(\alpha),\mathfrak{z}(\tau_1(\alpha)),\dots,\mathfrak{z}(\tau_r(\alpha)))\|\,d\alpha \\
& + \frac{1}{\Gamma(\beta)} \int_0^t (t-\alpha)^{\beta-1}\|R_{\mathfrak{z}}(t,\Theta)\| \\
& \times \left( \int_0^\alpha \|\mathfrak{g}(\alpha,\eta,\mathfrak{z}(\eta),\mathfrak{z}(\delta_1(\eta)),\dots,\mathfrak{z}(\delta_\kappa(\eta)))\|\,d\eta \right) d\alpha \\
& + \sum_{0<t_k<t} \|R_{\mathfrak{z}}(t,t_k)\|\,\|\mathfrak{I}_k(\mathfrak{z}(t_k))\|.
\end{aligned}
$$

Using the conditions $(H_1)$, $(H_2)$, $(H_4)$ and $(H_5)$, we get

$$
\begin{aligned}
\|\mathfrak{z}(t)\| \;\leq\; & \mathrm{Y}_0\|\mathfrak{z}_0\| + \mathrm{Y}_0\mathfrak{h}_1 + \mathrm{Y}_0\mathfrak{m}\mathfrak{l}\|\mathfrak{z}(t)\| \\
& + \mathrm{Y}_0\frac{1}{\Gamma(\beta)}\int_0^t (t-\alpha)^{\beta-1}\big(\mathfrak{f}_1(\|\mathfrak{z}(\alpha)\| + \|\mathfrak{z}(\tau_1(\alpha))\| + \dots \\
& + \|\mathfrak{z}(\tau_r(\alpha))\|)\big)d\alpha + \mathrm{Y}_0\frac{1}{\Gamma(\beta)}\int_0^t (t-\alpha)^{\beta-1}\big(\mathfrak{g}_1(\|\mathfrak{z}(\alpha)\| \\
& + \|\mathfrak{z}(\delta_1(\alpha))\| + \dots + \|\mathfrak{z}(\delta_\kappa(\alpha))\|)\big)d\alpha, \\
\leq\; & \mathrm{Y}_0\|\mathfrak{z}_0\| + \mathrm{Y}_0\mathfrak{h}_1 + \mathrm{Y}_0\mathfrak{m}\mathfrak{l}\|\mathfrak{z}(t)\| \\
& +\,_0D_t^{-\beta}[\mathfrak{f}_1(\|\mathfrak{z}(t)\| + \|\mathfrak{z}(\tau_1(t))\| + \dots + \|\mathfrak{z}(\tau_r(t))\|)]\mathrm{Y}_0 \\
& +\,_0D_t^{-\beta}[\mathfrak{g}_1(\|\mathfrak{z}(t)\| + \|\mathfrak{z}(\delta_1(t))\| + \dots + \|\mathfrak{z}(\delta_\kappa(t))\|)]\mathrm{Y}_0.
\end{aligned}
$$

There exists a non-negative function $M(t)$. We have

$$
\begin{aligned}
\|\mathfrak{z}(t)\| \;=\; & \mathrm{Y}_0\|\mathfrak{z}_0\| + \mathrm{Y}_0 k_5 + \mathrm{Y}_0\mathfrak{m}\mathfrak{l}\|\mathfrak{z}(t)\| \\
& +\,_0D_t^{-\beta}(\mathfrak{f}_1(\|\mathfrak{z}(t)\| + \|\mathfrak{z}(\tau_1(t))\| + \dots + \|\mathfrak{z}(\tau_r(t))\|))\mathrm{Y}_0 \\
& +\,_0D_t^{-\beta}(\mathfrak{g}_1(\|\mathfrak{z}(t)\| + \|\mathfrak{z}(\delta_1(t))\| + \dots + \|\mathfrak{z}(\delta_\kappa(t))\|))\mathrm{Y}_0 - M(t). \quad (4)
\end{aligned}
$$

Taking Laplace transformations of both sides of (4), we get

$$
\begin{aligned}
\|\mathfrak{z}(s)\| \;=\; & \frac{\mathrm{Y}_0\|\mathfrak{z}_0\|}{s} + \frac{\mathrm{Y}_0\mathfrak{h}_1}{s} + \mathrm{Y}_0\mathfrak{m}\mathfrak{l}\|\mathfrak{z}(s)\| + \mathrm{Y}_0\mathfrak{f}_1 s^{-\beta}(\|\mathfrak{z}(s)\| + \|\mathfrak{z}(\tau_1(s))\| + \dots \\
& + \|\mathfrak{z}(\tau_r(s))\|) + \mathrm{Y}_0\mathfrak{g}_1 s^{-\beta}(\|\mathfrak{z}(s)\| + \|\mathfrak{z}(\delta_1(s))\| + \dots + \|\mathfrak{z}(\delta_\kappa(s))\|) - M(s),
\end{aligned}
$$

$$
\begin{aligned}
\vartheta\left[\frac{s^\beta - \mu}{s^\beta}\right]\|\mathfrak{z}(s)\| \;=\; & \frac{1}{s}\Big[\mathrm{Y}_0\|\mathfrak{z}_0\| + \mathrm{Y}_0\mathfrak{h}_1 + \mathrm{Y}_0\mathfrak{f}_1 s^{1-\beta}(\|\mathfrak{z}(\tau_1(s))\| + \dots + \|\mathfrak{z}(\tau_r(s))\|) \\
& + \mathrm{Y}_0\mathfrak{g}_1 s^{1-\beta}(\|\mathfrak{z}(\delta_1(s))\| + \dots + \|\mathfrak{z}(\delta_\kappa(s))\|) - sM(s)\Big].
\end{aligned}
$$

Then,

$$
\begin{aligned}
\vartheta\|\mathfrak{z}(s)\| \;=\; & \frac{1}{s[s^\beta - \mu]}\Big[s^\beta \mathrm{Y}_0\|\mathfrak{z}_0\| + \mathrm{Y}_0\mathfrak{h}_1 s^\beta + \mathrm{Y}_0\mathfrak{f}_1 s(\|\mathfrak{z}(\tau_1(s))\| + \dots + \|\mathfrak{z}(\tau_r(s))\|) \\
& + \mathrm{Y}_0\mathfrak{g}_1 s[\|\mathfrak{z}(\delta_1(s))\| + \dots + \|\mathfrak{z}(\delta_\kappa(s))\|] - s^{\beta+1}M(s)\Big] \\
=\; & \frac{1}{s^\beta - \mu}\Big[s^{\beta-1}\mathrm{Y}_0[\|\mathfrak{z}_0\| + \mathfrak{h}_1] - s^\beta M(s) \\
& + \mathrm{Y}_0\mathfrak{f}_1 s^{\beta-\beta}(\|\mathfrak{z}(\tau_1(s))\| + \dots + \|\mathfrak{z}(\tau_r(s))\|) \\
& + \mathrm{Y}_0\mathfrak{g}_1 s^{\beta-\beta}(\|\mathfrak{z}(\delta_1(s))\| + \dots + \|\mathfrak{z}(\delta_\kappa(s))\|)\Big]. \quad (5)
\end{aligned}
$$

Taking Laplace inverse transformations on both sides of (5),

$$
\begin{aligned}
\vartheta\|\mathfrak{z}(t)\| &= \mathrm{Y}_0(\|\mathfrak{z}_0\| + \mathfrak{h}_1)E_{\beta,1}(\mu t^\beta) - M(t) * \left[t^{-1}E_{\beta,0}(\mu t^\beta)\right] \\
&\quad + \mathrm{Y}_0\mathfrak{f}_1\left[t^{\beta-1}E_{\beta,\beta}(\mu t^\beta)\right] * \left[\|\mathfrak{z}(\tau_1(t))\| + \ldots + \|\mathfrak{z}(\tau_r(t))\|\right] \\
&\quad + \mathrm{Y}_0\mathfrak{g}_1[t^{\beta-1}E_{\beta,\beta}(\mu t^\beta)] * \left[\|\mathfrak{z}(\delta_1(t))\| + \ldots + \|\mathfrak{z}(\delta_\kappa(t))\|\right], \\
&\leq \mathrm{Y}_0(\|\mathfrak{z}_0\| + \mathfrak{h}_1)E_{\beta,1}\left(\mu t^\beta\right),
\end{aligned}
$$

where * denotes the convolution operator; the terms involving with it are non-negative. Therefore, (3) has been achieved. Hence, from Definition 3, the solution of system (1) is Mittag–Leffler stable. □

In the case of the nonlocal term $\mathfrak{h}(\mathfrak{z}) = 0$, the initial condition of system (1) is reduced to $\mathfrak{z}(0) = \mathfrak{z}_0$, Then, the Mittag–Leffler stability results for this case can be achieved through the following corollary.

**Corollary 1.** *Let* $-A(t, \mathfrak{z}(t))$ *generate the bounded resolvent operator* $\|R_\mathfrak{z}(t, \Theta)\| \leq \mathrm{Y}e^{N(t-\Theta)}$ *with* $\mathrm{Y}_0 = \max\|R_\mathfrak{z}(t, \Theta)\|_Y$ *for all* $0 \leq \Theta \leq t \leq T$, $\mathfrak{z} \in \Omega$, *and the conditions* $(H_1)$–$(H_3)$, $(H_5)$ *hold. If there exist constants* $\mathfrak{f}_1$, $\mathfrak{g}_1$, *the mild solution of system* (1) *satisfies*

$$\|\mathfrak{z}(t)\| \leq (1/\vartheta)\mathrm{Y}_0\|\mathfrak{z}_0\|E_\beta(\mu t^\beta), \quad \forall t \in J,$$

*so the system* (1) *is Mittag–Leffler stable.*

In the case $\int_0^t \mathfrak{g}(t, \alpha, \mathfrak{z}(\alpha), \mathfrak{z}(\delta(\alpha)))\mathrm{d}\alpha = 0$ in (1), the system is reduced to an impulsive, fractional, nonlocal, quasilinear multi-delayed system of the form

$$
\begin{aligned}
D^\beta\mathfrak{z}(t) + A(t, \mathfrak{z}(t))\mathfrak{z}(t) &= \mathfrak{f}(t, \mathfrak{z}(t), \mathfrak{z}(\tau(t))), \\
\mathfrak{z}(0) + \mathfrak{h}(\mathfrak{z}) &= \mathfrak{z}_0, \\
\Delta\mathfrak{z}(t_k) &= \mathfrak{I}_k(\mathfrak{z}(t_k)), \quad k = 1, 2, \ldots, \mathfrak{m},
\end{aligned}
\tag{6}
$$

where $t \in J$. Then, the stability of (6) can be stated as follows:

**Corollary 2.** *Let* $-A(t, \mathfrak{z}(t))$ *generate the bounded resolvent operator* $\|R_\mathfrak{z}(t, \Theta)\| \leq \mathrm{Y}e^{N(t-\Theta)}$ *with* $\mathrm{Y}_0 = \max\|R_\mathfrak{z}(t, \Theta)\|_Y$ *for all* $0 \leq \Theta \leq t \leq T$, $\mathfrak{z} \in \Omega$, *and the conditions* $(H_1)$, $(H_3)$–$(H_5)$ *hold. If there exist constants* $\mathfrak{f}_1$, $\mathfrak{h}_1$, *the mild solution of system* (6) *satisfies*

$$\|\mathfrak{z}(t)\| \leq (1/\vartheta)\mathrm{Y}_0([\|\mathfrak{z}_0\| + \mathfrak{h}_1)E_\beta\left(\frac{\mathrm{Y}_0\mathfrak{f}_1}{1 - \mathrm{Y}_0\mathfrak{m}\mathfrak{l}}t^\beta\right), \quad \forall t \in J,$$

*so the system* (6) *is Mittag–Leffler stable.*

## 4. Application

Consider the fractional-order, nonlocal, impulsive, integro-differential systems with multiple delays of the form

$$
\begin{aligned}
\frac{\partial^\beta\mathfrak{z}(x,t)}{\partial t^\beta} + a(x, t, \mathfrak{z}(x,t))\frac{\partial^2\mathfrak{z}(x,t)}{\partial x^2} &= x\arctan\varphi_p(x, t, \mathfrak{z}) + \int_0^t e^{-\varphi_q(x,s,\mathfrak{z})}ds, \tag{7} \\
\mathfrak{z}(x, 0) + \sum_{k=1}^{\mathfrak{m}} c_k\mathfrak{z}(x, t_k) &= \mathfrak{z}_0(x), \ x \in [0, \pi], \\
\mathfrak{z}(0, t) = \mathfrak{z}(\pi, t) &= 0, \ t \in J, \\
\Delta\mathfrak{z}(t_k, x) &= \frac{\mathfrak{z}(t_k, x)}{2 + \mathfrak{z}(t_k, x)}, \ x \in (0, 1), \ k = 1, \ldots, \mathfrak{m},
\end{aligned}
$$

where $0 < \beta \leq 1$, $0 < t_1 < \ldots < t_{\mathfrak{m}} < T$. Let $X = L^2[0, \pi]$, $PC = PC(J, S_\delta)$, $S_\delta = \{y \in L^2[0, \pi]\colon \|y\| \leq \delta\}$. First, we prove that $-A(t, \mathfrak{z}(t))$ generates the bounded resolvent operator $R_{\mathfrak{z}}(t, \Theta)$ with the help of the following analysis. Let $a(x, t, \mathfrak{z}(x, t))$ be continuous; define $A(t, .) : X \longrightarrow X$ by $(A(t, .)w)(x) = a(x, t, \mathfrak{z}(x, t))w''$ with domain $D(A(t, .)) = \{w \in X\colon w, w'$ being absolutely continuous, $w'' \in X; w(0) = w(\pi) = 0\}$ is dense in the $X$ and independent of $t$. Then,

$$A(t, \mathfrak{z})w = \sum_{n=1}^{\infty} n^2(w, w_n), w \in D(A), \tag{8}$$

where $(., .)$ is the inner product in $L^2[0, \pi]$, $w_n = Z_n \circ \mathfrak{z}$ is the orthogonal set of eigenvectors in $A(t, \mathfrak{z})$ and $Z_n(t, s) = \sqrt{\frac{2}{\pi}} \sin n(t - s)^\beta$, $0 < \beta \leq 1$, $0 \leq s \leq t \leq a$, $n = 1, 2, \ldots$

Then, the operator $[A(t, .) + \lambda^\beta I]^{-1}$ exits in $L(X)$ for any $\lambda$ with $\mathrm{Re}\lambda \leq 0$ and

$$\|[A(t, .) + \lambda^\beta I]^{-1}\| \leq \frac{C_\alpha}{|\lambda| + 1}, \ t \in J. \tag{9}$$

Additionally, there exist constants $\nu \in (0, 1]$ and $C_\beta$ such that

$$\|[A(t_1, .) - A(t_2, .)]A^{-1}(s, .)\| \leq C_\beta |t_1 - t_2|^\eta, \ t_1, t_2, s \in J. \tag{10}$$

Under the conditions (8)–(10), each operator $-A(s, .)$, $s \in J$ generates an evolution operator $\exp(-t^\alpha A(s, .))$ for $t > 0$, and there exists a constant $C_\alpha$ such that

$$\|A^n(s, .)\exp(-t^\beta A(s, .))\| \leq \frac{C_\beta}{t^n}, \quad \forall\, n = 0, 1, \ t > 0, \ s \in J.$$

Therefore, it can be concluded that the evolution operator of the $(\beta, \mathfrak{z})$ resolvent family has the form

$$R_{(\beta, \mathfrak{z})}(t, s)w = \sum_{n=1}^{\infty} \exp[-n^2(t - s)^\beta](w, w_n)w_n, \ w \in X.$$

From (7), the functions $\mathfrak{f}(\cdot)$, $\mathfrak{g}(\cdot)$ are given by $\mathfrak{f}(t, \mathfrak{z}(\beta(t))) = x\arctan \varphi_p(x, t, \mathfrak{z})$ and $\mathfrak{g}(t, s, \mathfrak{z}(\gamma(t))) = e^{-\varphi_q(x, s, \mathfrak{z})}$, which satisfies the assumptions $(H_1)$–$(H_3)$ for $\varphi_\eta(x, s, \mathfrak{z}) = (\mathfrak{z}(x, \sin t), \mathfrak{z}(x, (\sin t)/2), \ldots, \mathfrak{z}(x, (\sin t)/\eta))$ and $\beta_\tau(t) = \gamma_\tau(t) = (\sin t)/\tau$, $\tau = 1, \ldots, \eta$, $\eta = \max(r, k)$.

Additionally, from the nonlocal (function) initial condition, $\mathfrak{h}(\mathfrak{z}(., t)) = \sum_{k=1}^{\mathfrak{m}} c_k \mathfrak{z}(., t_k)$ will satisfy Assumption $(H_4)$ with $\sum_{k=1}^{\mathfrak{m}} c_k = \mathfrak{h}_1$. Further, the at impulse moments $\mathfrak{I}_k(\mathfrak{z}(t_k)) = \frac{\mathfrak{z}(t_k, x)}{2 + \mathfrak{z}(t_k, x)}$ satisfies Assumption $(H_5)$ with $\mathfrak{l} = \frac{1}{2}$.

Thus, Assumptions $(H_1)$–$(H_5)$ (all) are satisfied, and it is possible to choose the constants in Theorem 2, which satisfy the required stability condition (3). Hence, by Definition 3, the considered system (7) is MLS on $J$.

## 5. Conclusions

The Mittag–Leffler stability results for a class of fractional-order, quasilinear, impulsive, integro-differential systems with multiple delays has been investigated. Based on the contraction mapping principle, the existence and uniqueness of a solution for the FOS was achieved. Then, novel conditions for MLS of the considered system were derived by using well known mathematical techniques, and further, some corollaries were proposed for the cases of initial conditions without a nonlocal term and an FOS in the absence of

an integro-differential part. At last, the presented results were verified with an example, which illustrated the applications.

**Author Contributions:** All authors contributed equally to this article. All authors have read and agreed to the published version of the manuscript.

**Funding:** This research received no external funding.

**Conflicts of Interest:** The authors declare no conflict of interest.

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
