# Peer review of "Stability of Fractional-Order Quasi-Linear Impulsive Integro-Differential Systems with Multiple Delays"

_axioms, doi:10.3390/axioms11070308_

Round 1

Reviewer 1 Report

All my comments are in a pdf file, where I have marked the places in green and the comments in red.

In addition, the following questions and comments arise for me:

1.the work is not prepared in the appropriate template,

2. item 29 in the literature is not quoted in the work,

3. for me it was a terrible idea to choose a font in a mathematical setting - is it a standard in this type of work (I doubt it) or an idea of the authors? Maybe you have ideas about changing the font to the standard one?

4. Can the authors provide an example of an engineering problem (technical, physical) which is described by the equation considered in the study?

Reviewer 2 Report

See enclosed report

Reviewer 3 Report

Axioms-1767443

Authors: K. Mathiyalagan, Shengda Zeng, Mehmet Yavuz

This paper studies the Mittag-Leffler stability results for a class of fractional-order quasilinear impulsive integro-differential systems with multiple delays. The existence and uniqueness of mild solutions for the considered system is discussed using contraction mapping theorem. Then, novel conditions for Mittag-Leffler stability of the considered system are established by using well known mathematical techniques and some corollaries are deduced giving new results. The presented results are verified with an example, which illustrates the applications.

The paper is interesting, well organized and has an adequate set of references. Yet some improvements in the English writing  are in order. For instance, the first sentence in the introduction is somehow confusing and needs rewriting. The third last sentence in the introduction as 2 verbs. Generally, you can find throughout the text some typos and other mistakes that must be corrected. The acronym FOS, though easily understanded, must be defined.
